# Longitudinal Predictors of Self-Regulation at School Entry: Findings from the All Our Families Cohort

**DOI:** 10.3390/children7100186

**Published:** 2020-10-16

**Authors:** Erin Hetherington, Sheila McDonald, Nicole Racine, Suzanne Tough

**Affiliations:** 1Department of Community Health Sciences, Cumming School of Medicine, University of Calgary 3330 Hospital Drive NW, Calgary, AB T2N 4N1, Canada; elhether@ucalgary.ca (E.H.); Sheila.McDonald@albertahealthservices.ca (S.M.); 2Department of Pediatrics, Cumming School of Medicine, University of Calgary, 28 Oki Drive NW, Calgary, AB T3B 6A8, Canada; 3Department of Psychology, University of Calgary, 2500 University Dr. N.W., Calgary, AB T2N 1N4, Canada; nicole.racine2@ucalgary.ca

**Keywords:** child behavior, child development, self-regulation, parenting, screen time, longitudinal cohort

## Abstract

Self-regulation is the ability to manage emotions, modulate behaviors, and focus attention. This critical skill begins to develop in infancy, improves substantially in early childhood and continues through adolescence, and has been linked to long-term health and well-being. The objectives of this study were to determine risk factors and moderators associated with the three elements of self-regulation (i.e., inattention, emotional control, or behavioral control) as well as overall self-regulation, among children at age 5. Participants were mother–child dyads from the All Our Families study (*n* = 1644). Self-regulation was assessed at age 5. Risk factors included income, maternal mental health, child sex, and screen time, and potential moderation by parenting and childcare. Adjusted odds ratios of children being at risk for poor self were estimated using multivariable logistic regression. Twenty-one percent of children had poor self-regulation skills. Risk factors for poor self-regulation included lower income, maternal mental health difficulties, and male sex. Childcare and poor parenting did not moderate these associations and hostile and ineffective parenting was independently associated with poor self-regulation. Excess screen time (>1 h per day) was associated with poor self-regulation. Self-regulation involves a complex and overlapping set of skills and risk factors that operate differently on different elements. Parenting and participation in childcare do not appear to moderate the associations between lower income, maternal mental health, male sex, and screen time with child self-regulation.

## 1. Introduction

The first five years of life have been identified as critical for healthy development, as they set the stage health and well-being across the life course [1]. A key developmental skill acquired in these early years is the capacity for self-regulation, which is defined as the ability to monitor and control emotions, attention, and interactions with others and the environment [2,3]. These foundational skills, which develop from infancy and mature through adolescence, are critical to the development of positive relationships with family members, teachers, and peers [2,4]. Longitudinal studies of self-regulation show that preschoolers who could wait for two marshmallows instead of eating one immediately had higher educational attainment, lower use of addictive substances, and lower rates of obesity later in life [5]. Other studies have found that poor self-regulation in the early years can increase the risk for poor academic performance, lower social competence, smoking, unemployment and relational stress [6,7]. Taken together, understanding the factors that predict self-regulation has the potential to mitigate poor outcomes associated with poor self-regulation.

While there continues to be considerable variation in what constitutes the specific elements of self-regulation, recent large-scale reviews have agreed that self-regulation includes emotional regulation, attention regulation, as well as the ability to control behaviors [2,4]. Emotional regulation refers to strategies and skills that can manage, enhance or inhibit emotional experiences and expressions [3]. These may rely on external aids (comfort from others) or internal (taking deep breaths). Attention regulation involves focusing on tasks or instructions, including listening for sustained periods [4]. Finally, the ability to control behaviors includes reacting appropriately in social situations (e.g., waiting to take a turn) and inhibiting inappropriate responses (e.g., hitting another child) [4].

Research to date has focused on independent risk or protective factors for the development of self-regulation, or on specific elements of self-regulation (attention, emotion regulation and behavioral control), but not overall self-regulation as a broader construct. Common predictors of poor self-regulation (or specific elements) include family income, maternal mental health, child sex, parenting, and, to a lesser extent, childcare and screen time [2,3,8,9,10]. What is not as well understood, is how these factors may work together, including exacerbation or mitigation effects among predictors on child self-regulation. The current study sought to address this research gap.

### 1.1. Risk Factors for Self-Regulation

There are several identified risk factors for self-regulation in the literature. For example, lower income remains a consistent predictor of self-regulation skills. Some studies suggest that economically disadvantaged children may have access to fewer resources and opportunities to engage in enrichment activities (e.g., dance classes or sports) that promote self-regulation skills [8,11,12]. Additional research suggests that living in stressful or chaotic environments could impact self-regulation [13].

Maternal depression has also been associated with the development of sub-optimal emotion regulation skills in children [14,15]. Mothers with mental health challenges may be less responsive to their children, and may not model appropriate positive social and emotional behaviors [16]. One large population based study in Canada found that children of mothers with depression were less able to focus attention [17]. There is little information regarding the impact of maternal anxiety on overall self-regulation, or if the negative association between maternal mental health and self-regulation can be moderated by external factors.

Gender has also been identified as playing a role in the development of self-regulation. Many studies note that self-regulation skills seem to develop later in boys with marked differences emerging in early childhood and persisting through middle childhood, but evidence on moderators is mixed [3,8,18].

Finally, there is emerging evidence that screen time may impact self-regulation. Two large scale studies found that increased television viewing was related to lower self-regulation skills [9,10]. Both of these studies measured screen time simultaneously with self-regulation and used short scales (five or six items) to measure self-regulation. Longitudinal studies with robust measures of self-regulation are needed to better understand this association.

### 1.2. Potential Moderators

This study focuses on two potential moderators of the association between risk factors listed above and child self-regulation: group-based childcare and parenting (Figure 1). Previous studies suggest that childcare is associated with better behavioral outcomes, such as decreased physical aggression, lower hyperactivity, and higher prosocial behavior [17,19] Evidence from Australia suggests that participation in childcare improved emotional regulation in children [20]. Therefore, while evidence suggests that childcare may be beneficial for distinct elements, there is little evidence looking at the role of childcare at the larger construct of self-regulation.

Previous research has examined the role of childcare as a moderator between low income and academic achievement, suggesting a potential effect on attention and cognitive skills [11,21]. However, it is not known if this moderation extends to other elements of self-regulation. The hypothesis is that childcare provides opportunities for children to interact in a structured environment with skilled providers, which facilitates development of emotional regulation and behavioral control. Also, there is evidence that childcare can moderate the negative association between maternal mental health and behavioral control, however, the potential moderating role of childcare for overall self-regulation has yet to be examined [17].

Parenting is thought to influence self-regulation through a modeling, interaction, and the emotional environment in the home [22]. Parenting behaviors such as high levels of negative reinforcement and low levels of positive interaction are consistently associated with lower emotion regulation and greater behavioral challenges both in cross-sectional and longitudinal studies of children [23,24,25]. Thus, it is possible that parenting practices may play a role in attenuating or exacerbating the association between male sex or screen time and children’s self-regulation.

In terms of parenting moderating the association between child sex and self-regulation, data from the Longitudinal Study of Australian Children found that harsh parenting did not moderate this association at ages 2 and 3 years [25]. Similar results were found from the National Longitudinal Survey of Youth among older children aged 8 and 9 years [23]. However, other studies have shown that boys’ emotion regulation and behavior are more susceptible to parenting behavior than girls [26,27]. Previous research has focused on how different parenting practices are associated with increased or decreased screen time in young children [28]. However, there is little information on whether parenting may moderate the association between screen time and self-regulation. For example, the potential negative influence of screen time on children’s self-regulatory abilities may be negated by supportive, positive, and engaging parenting.

Overall the literature points to some clear risk factors for elements of poor self-regulation, including lower income, maternal mental health, child sex, and screen time. However, how these risk factors might be moderated by childcare and parenting is less clear. The purpose of this study is to identify whether modifiable factors (such as childcare and parenting behavior) moderate known risk factors including lower income, maternal mental health and child sex on child self-regulation. Specifically, we hypothesized the negative associations between child self-regulation and both lower income and maternal mental health would be moderated by participation in childcare outside the home. We also hypothesized that the associations between child self-regulation and both child sex and screen time would be moderated by parenting behavior. Because overall self-regulation is comprised of emotional, behavioral and attention elements, a secondary purpose was to understand the above associations on any or all elements of self-regulation.

## 2. Materials and Methods

### 2.1. Participants

This study is a secondary data analysis using data from mothers and children participating in the All Our Families longitudinal pregnancy cohort (formerly All Our Babies) [29]. Briefly, women who were less than 25 weeks gestation, at least 18 years of age, accessing prenatal care in Calgary, and able to complete a questionnaire in English were recruited between May 2008 and May 2011. Questionnaires were given out twice during pregnancy, at 4 months postpartum, and at 1, 2, 3, and 5 years. Response rates varied between 69% and 99% depending on data wave and eligibility to participate in that time period [29]. The current study focuses on self-regulation in children at age 5. Because we were interested in predictors of self-regulation among typically developing children, those with a diagnosed developmental delay (autism, cerebral palsy, etc.) were excluded (*n* = 26). For the current analysis, we included women who responded to both the three-year and five-year questionnaire (completed in 2015 and 2017; *n* = 1688), which is a composite response rate of 60% of 2819 eligible participants. We excluded 18 participants with incomplete information on the outcome variable, resulting in a final sample of 1644. See Appendix A for study flow chart. This study received ethical approval from the University of Calgary Conjoint Health Research Ethics Board. All subjects gave their informed consent for inclusion before they participated in the study. Our study received ethical approval from the Conjoint Health Research Ethics Board of the University of Calgary REB13-0868.

### 2.2. Measures

The outcome measure was self-regulation skills at age 5, which was measured by parent report on the Behavior Assessment System for Children (BASC-2). The BASC-2 is a comprehensive behavioral assessment including 134 questions which are summarized into 23 scales. Parents rate frequency of behaviors and raw scores are converted to T-scores with mean = 50, Standard Deviation (SD) = 10. Scores between 60 and 69 are considered “at risk” and scores of 70 are “clinically significant”. Consistent with the theoretical literature, we operationalized poor self-regulation as children who scored “at risk” or above on one or more of the inattention, emotional control, or executive function scales. The inattention scale covered concepts about listening and focusing attention. The emotional control scale included concepts such as losing one’s temper and quick mood changes. The executive function scale included a high number of behavior control items such as hitting other children or waiting one’s turn. The BASC-2 is appropriate for use in the general population, has strong psychometric properties, and the scales correspond to the theoretical literature on how to measure self-regulation [4,8].

Risk factors included income, maternal mental health, child sex, screen time, childcare, and parenting, and were measured at age 3. Income was self-reported household income. Maternal depression was measured using the 20-item Center for Epidemiology Studies–Depression (CES-D) scale. A score of 16 or more is considered clinically relevant and Cronbach’s alpha in this sample was 0.89 [30]. Maternal anxiety was measured using the 20 item Speilberger State Anxiety Scale (SSAI). A score of 40 or more considered clinically significant, and Cronbach’s alpha is this sample was 0.93 [31]. Child sex was reported by mothers at birth. Screen time was measured by maternal report of how much time they estimated their child spent watching television, movies, or playing videogames per day. Responses ranged from less than 1 h to over 5 h and were dichotomized according to national guidelines for this age group (1 h or less compared to more than 1 h per day) [32]. Children were considered to be in childcare if they spent more than 10 h a week outside the home in a group-based childcare [21]. Parenting was measured using the National Longitudinal Study of Children and Youth (NLSCY) parenting scales which measure positive parenting (praise and support) and hostile/ineffective parenting (anger and repeated commands). Scales showed adequate internal consistency with Cronbach’s alphas for each scale of 0.75. Because parenting scales have no validated cut-offs, we used a 1 standard deviation cut-off to indicate low levels of positive parenting and high levels of hostile/ineffective parenting. Control variables included child age in months and maternal age in years.

### 2.3. Analysis

Descriptive statistics were provided for all variables. Among the 1644 included participants, there was less than 1% missing data on any given variable, thus we conducted a complete case analysis. To understand the components of self-regulation, the number of children scoring at risk on one or more of the inattention, emotional control, or executive function scales was calculated. The children scoring at risk on one, two, or more of each of the scales was depicted graphically. To assess effect modification, interaction terms were created for the following variables: income and childcare, maternal depression and childcare, maternal anxiety and childcare, parenting and child sex, and parenting and screen time. We developed five models to estimate Odds Ratios (OR) and 95% Confidence Intervals (CI) between risk factors and self-regulation. Model 1 estimated odds of poor self-regulation based on scoring “at risk” on one or more of the scales. Model 2 estimated odds of poor self-regulation based on scoring “at risk” on all of the scales. Models 3, 4, and 5 estimated odds of poor inattention, emotional control, or behavioral control, respectively. Non-significant interaction terms (<0.05) were dropped. All models were adjusted for all other variables in the model as well as maternal age in years and child age in months. A sensitivity analysis was conducted to ensure dichotomization did not result in loss of information by running all predictors with continuous values. A robustness analysis was conducted using 20 h of childcare per week. All analyses were completed using STATA v.16.

## 3. Results

Descriptive statistics for participants are presented in Table 1. Characteristics of poor self-regulation by constituent elements are shown in Figure 2. Of the 1644 children, 354 (21.5%) were at risk on any element of poor self-regulation, and only 64 were at risk for all elements (3.9%). Thirteen percent of children were at risk for poor emotional self-control (*n* = 221), 11.0% were at risk of poor behavior control (*n* = 181), and 10.9% were at risk of inattention (*n* = 179). Among those at risk on at least one scale (*n* = 354) approximately 54% of children were only at risk on one scale (*n* = 191), 28% were at risk on two scales (*n* = 99), and 18% on all three scales (*n* = 64). The overlap between children with “at risk” levels of inattention, poor emotional control, or poor behavior control can be seen in Figure 2.

Adjusted odds ratios for any element, all elements, and each element of poor self-regulation can be seen in Table 2. None of the potential moderators were statistically significant, and therefore are not included in the results. Increasing income was associated with decreased odds of poor self-regulation in Models 1, 3, and 5. Maternal mental health at 3 years (depression or anxiety) had consistently elevated point estimates across all models. For Models 1, 3, and 5, maternal anxiety at 3 years was associated with increased odds for elements of self-regulation (any, inattention, and behavior control), whereas maternal depression at age 3 was associated with increased odds for being at risk on all elements or on emotional self-control only (Models 2 and 4). Group childcare did not moderate either of these predictors in any of the models, but was independently associated with increased odds poor self-regulation in Model 3 (inattention).

Male children had increased odds of poor self-regulation across all models, except emotional self-control (Model 4). Elevated screen time was associated with increased odds of poor self-regulation in Model 1 (any element) and Model 3 (inattention). Neither hostile/ineffective parenting or positive parenting moderated either of these associations. However, high levels of hostile/ineffective parenting were associated with increased odds of poor self-regulation across all models.

Sensitivity analyses showing the models using continuous predictors can be seen in the Appendix A with consistent results. In these models, one additional hour of screen time per day is associated with a 1.23 increased odds of any element of poor self-regulation (Model 1; 95% CI 1.03, 1.47), and a 1.42 increased odds of inattention (Model 3; 95% CI 1.13, 1.79). The robustness analysis increasing the hours in childcare from 10 to 20 per week did not result in any meaningful changes.

## 4. Discussion

Our findings revealed that there is considerable overlap between different elements of self-regulation, as measured in the current study, including attention, emotional control, and behavioral control. This is consistent with the theoretical literature [2,4] and emphasizes that children may have challenges in one or multiple areas, possibly indicating increasing levels of severity of self-regulation challenges. Contrary to expectations, neither childcare nor poor parenting moderated the associations between predictors at age 3 (income, maternal mental health, male sex, or screen time) and self-regulation at age 5. Maternal mental health (maternal anxiety or depression), male sex, and high levels of ineffective/hostile parenting were consistently associated with elements of poor self-regulation across all models. Higher income was generally associated with lower odds of poor self-regulation, whereas participation in childcare and screen time associations varied.

Consistent with previous studies, our results showed that higher income was associated with decreased odds of poor-self regulation [8,12,18]. However, contrary to expectations, participation in childcare did not moderate this association. We posit two possible reasons for this lack of moderation. First, our study measured time in group child care as opposed to quality of child care and quality has been shown to be a better predictor of child outcomes [20]. Second, while only 35% of children were in childcare for more than 10 h per week at age 3, by age 5, almost all children were either in group childcare, preschool or kindergarten at least 10 h a week. This may have meant that a possible effect of childcare at age 3 was masked by increasing participation in childcare at the time of the outcome at age 5.

Our study is consistent with previous studies showing an association between maternal depression and lower emotion regulation in children [14,15]. Our study adds to our understanding of maternal mental health’s role by demonstrating a consistent association between maternal anxiety and other elements of self-regulation, including inattention and behavior control. Because mental health symptoms often co-occur, it is important to recognize that overall maternal mental health (characterized by either depressive or anxiety symptoms) may be an important predictor of child self-regulation. However different symptoms may operate differently on different aspects of self-regulation with depressive symptoms impacting emotion regulation (Model 4) and anxiety symptoms impacting inattention and behavior control (Models 3 and 5).

There was an unexpected increased odds of inattention at age 5 with participation in childcare at age 3 (Model 3) and non-statistically significant elevated odds of childcare influencing behavior control and any or all elements of self-regulation (Models 1, 2, and 5). Previous work suggests that childcare instability is associate with more hyperactivity and inattention [17]. As noted above, we were not able to account for quality or consistency of childcare which may partially explain this unexpected result.

Our study adds to the growing literature regarding concerns of excess screen time and child development [33,34]. Our results show a modest, but statistically significant relationship between screen time and problems with any element self-regulation (Model 1; adjusted odd ratios (AOR): 1.34, 95% CI 1.03, 1.73). This result was consistent whether screen time was categorized according to guidelines (1 h per day), but also showed a dose-response relationship with increasing odds for every additional hour of screen time. Our study adds to our understanding of this relationship by showing that the association between screen time and self-regulation is predominantly driven by the association with inattention, as opposed to other elements of self-regulation (AOR for inattention: 1.69, 95% CI 1.20, 2.38). Although our study did not measure the content of screen time viewing, previous research suggests that entertainment related television content is associated with attentional problems, but not educational content, and that children in this age group are much more likely to watch non-educational content [35]. Screen time among children exceeded guidelines before the global Covid-19 pandemic, and is expected to increase with physical distancing measures, online learning, and parents working from home [36,37]. Child health advocates caution about increased sedentary behavior and physical health impacts during the pandemic, but more research is needed into possible developmental challenges associated with excess screen time [38].

We found no evidence for moderation of either child sex or screen time by parenting. However, our results are consistent with other cross-sectional studies linking hostile/ineffective parenting as an independent predictor of poor self-regulation [39,40]. Specifically, hostile/ineffective parenting was associated with an adjusted odds ratio of 3.32 (95% CI 2.44, 4.53) of overall poor self-regulation. There was no independent effect of positive parenting on overall poor self-regulation. While the relationship between parenting and child behavior is likely in part bidirectional [41], our longitudinal results suggest that hostile parenting behaviors as early as age 3 may have harmful effects on the development of self-regulatory skills at age 5. While our study did not control for prior self-regulation, a study by Colman et al. noted that parenting practices were still associated with later self-regulation even when controlling for earlier self-regulating skills [23].

Our study’s strengths include a complex measure of self-regulation as captured by three scales on the BASC-2, which reflect the key components of self-regulation including emotional regulation, behavior control, and attention. While direct observation of self-regulation skills by an independent observer would be a gold-standard for measurement, the size of our sample makes direct observation unfeasible. Our study had a large sample size and longitudinal design which allowed us to examine risk factors which were measured at a timepoint prior to the outcome.

Our study also has several limitations. First, we had a high number of non-responders, which resulted in a collective response rate over two waves of data collection at 5 years of 60%. Responders were more socioeconomically advantaged (higher income and education) and older than non-responders [29]. This limits the generalizability of our results, and our findings may not be applicable to more disadvantaged groups. As we expect self-regulation challenges to be higher in more disadvantaged groups, this may mean our prevalence of self-regulation skills is underestimated and our associations with predictors to be biased towards the null. However, research from other longitudinal child behavior studies suggests that adjusted analysis minimizes the magnitude of this type of bias [42]. Second, measures of maternal mental, parenting, and child behavior were all reported by mothers which could result in reporting bias [43]. However, previous research attempting to quantify reporting bias through multiple observers have found that associations remain after accounting for shared variance [44].

The goal of our study was to assess known risk factors and potential moderators, and we did not attempt to assess potential mediations. Future studies could identify potential pathways and mediation mechanism. As our study is a secondary data analysis, we were not able to assess all possible factors. Previous research suggests sleep is an important predictor of self-regulation, and future studies should include a measure of sleep [45].

## 5. Conclusions

While our findings indicate that multiple factors contribute to the development of self-regulation in early childhood, the most consistent predictors of self-regulation challenges at age 5 were lower income, male sex, and inconsistent/hostile parenting at age 3. We did not find evidence for effect modification in our models. Parenting programs based on improving parenting practices that focus on positive relational interactions have been shown to improve child behavior, and can be effective at early ages [46]. With increasing levels of screen time among young children, our findings regarding screen time’s negative association with attentional elements of self-regulation warrants additional research. This point is of particular concern among rising screen time due to the Covid-19 pandemic. Our findings suggest that risk factors for poor self-regulation in children are evident much earlier than school entry and provide an opportunity for early identification of children at risk.

## Figures and Tables

**Figure 1 children-07-00186-f001:**
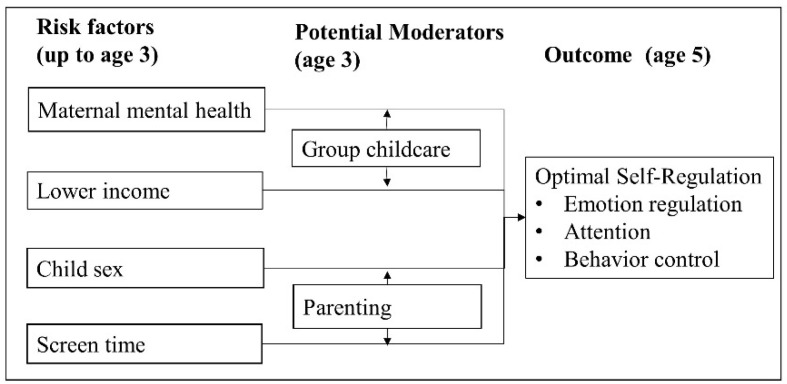
Risk factors and potential moderators of self-regulation. Risk factors (maternal mental health, lower income, child sex, and screen time) were measured at age 3 (or at birth in the case of child sex). Group childcare (at age 3) is hypothesized to moderate the association between maternal mental health or lower income and self-regulation. Parenting behaviors (at age 3) are hypothesized to moderate the association between child sex or screen time and self-regulation.

**Figure 2 children-07-00186-f002:**
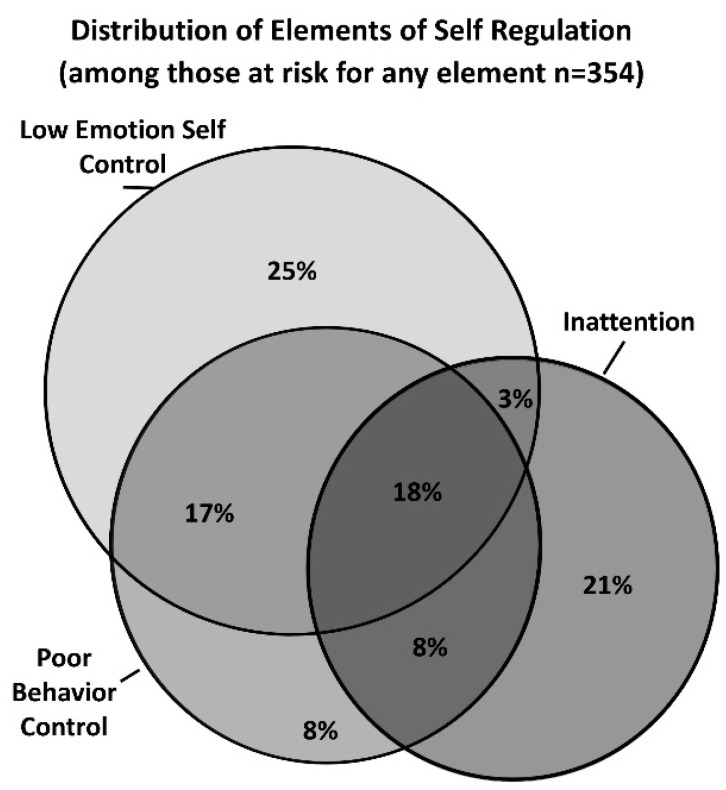
Elements of self-regulation among children who scored at risk on emotional self control, behavior control, or attention scales of the Behavior Assessment System for Children (BASC-2; *N* = 354).

**Table 1 children-07-00186-t001:** Characteristics of children and families.

	*N* = 1644 (%)
Maternal age in years—mean (SD)	36.49 (4.3)
Child age in months—mean (SD)	61.52 (3.0)
Maternal education	
high school or less	96 (5.8)
partial or complete university/college or trade	1253 (76.2)
partial or complete graduate education	295 (17.9)
Male sex	857 (52.3)
Maternal depression @ 3 years	191 (11.6)
Maternal anxiety @ 3 years	235 (14.4)
Household income <$50,000	99 (6.1)
$50,000–99,999	470 (28.8)
$100,000–149,999	523 (32.1)
$150,000 and above	539 (33.0)
Low positive parenting @ 3 years	204 (12.4)
High hostile parenting @ 3 years	252 (15.4)
Childcare 10+ h/wk @ 3 years	584 (35.5)
More than 1 h screen time/day @ 3 years	855 (52.0)

Maternal anxiety measured as scoring over 40 on the SSAI. Maternal depression measured as scoring over 16 on the CES-D. Parenting measured as 1 SD above or below mean on NLSCY scales. SD: Standard Deviation; SSAI: Speilberger State Anxiety Scale; Hrs/wk: hours per week; CES-D: Center for Epidemiology Studies–Depression; NLSCY: National Longitudinal Study of Children and Youth.

**Table 2 children-07-00186-t002:** Adjusted odd ratios (AOR) for poor self-regulation.

Models	Model 1: Any Element of Poor Self-Regulation		Model 2: All Elements of Poor Self-Regulation		Model 3: Inattention		Model 4: Low Emotional Control		Model 5: Low Behavioral Control	
	**AOR**	**95% CI**	**AOR**	**95% CI**	AOR	95% CI	AOR	95% CI	AOR	95% CI
Increasing income	0.88	(0.82, 0.95)	0.98	(0.84, 1.14)	0.87	(0.79, 0.96)	0.94	(0.86, 1.02)	0.92	(0.83, 1.01)
Maternal anxiety @ 3 years	**1.72**	**(1.17, 2.53)**	1.41	(0.68, 2.95)	**1.89**	**(1.18, 3.02)**	1.16	(0.73, 1.83)	**2.13**	**(1.34, 3.39)**
Maternal depression @ 3 years	1.28	(0.84, 1.96)	1.71	(0.79, 3.68)	1.16	(0.69, 1.95)	**1.78**	**(1.10, 2.86)**	1.25	(0.75, 2.08)
Childcare (10+ h/week)	1.22	(0.93, 1.59)	1.65	(0.95, 2.85)	**1.82**	**(1.29, 2.57)**	0.94	(0.68, 1.30)	1.21	(0.85, 1.72)
Male child	**1.30**	**(1.01, 1.68)**	**2.20**	**(1.25, 3.88)**	**1.86**	**(1.32, 2.62)**	1.09	(0.81, 1.47)	**1.60**	**(1.14, 2.25)**
Screen time (>1 h/day)	**1.34**	**(1.03, 1.73)**	1.54	(0.89, 2.68)	**1.69**	**(1.20, 2.38)**	1.14	(0.84, 1.54)	**1.38**	**(0.98, 1.95)**
High hostile/ineffective parenting	**3.32**	**(2.44, 4.52)**	**2.72**	**(1.52, 4.87)**	**2.82**	**(1.92, 4.13)**	**2.49**	**(1.75, 3.54)**	**3.95**	**(2.73, 5.71)**
Low positive parenting	0.84	(0.58, 1.23)	1.22	(0.61, 2.42)	0.79	(0.48, 1.28)	1.09	(0.71, 1.68)	0.83	(0.51, 1.35)

AOR: Adjusted odds ratio—adjusted for all variables in the table and maternal age, maternal education, and child age. CI: Confidence Interval. Original models included the following interaction terms: income*hildcare, maternal anxiety*childcare, maternal depression*childcare, male child*hostile parenting, male child*positive parenting, screen time*hostile parenting, screen time*positive parenting. All interaction terms were dropped because they were not statistically significant (0.05) All statistically significant predictors (at 0.05) are bolded, The * refers to the mathematical symbol for multiplication.

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
