# Peer review of "Longitudinal Predictors of Self-Regulation at School Entry: Findings from the All Our Families Cohort"

_children, 2020, doi:10.3390/children7100186_

Round 1

Reviewer 1 Report

  1. Line 13: Abstract: Do not use the same term “regulate” in the definition of that term.  Doing so is called a “circular definition”.  Use a word such as “effortful control” or “modulate” behaviors.
  2. Line 4: Add a qualifying word to “develops in early childhood”:  SR begins to develop in infancy, improves substantially in early childhood and continues so throughout childhood.
  3. Line 16: “overall poor self-regulation, and elements of self-regulation…”:

This doesn’t seem to fit here and “elements” need to be clarified.    Perhaps reword the 2nd sentence so it’s clear that you mean the three elements (named in 1st sentence).  For example:  The objectives (plural?) ….associated with these three elements of SR, as well as overall poor SR…

  1. Risk factors: why was temperament, a well-recognized component of SR, not included? See Rothbart; see Duckworth.
  2. Line 53: use behavioral control
  3. Line 63: delete “which”
  4. Line 68: delete “and”
  5. Line 69: do not use two “on” in the same sentence. Reword this sentence.
  6. Line 83: specify what type of childcare, e.g. high quality?
  7. Line 103: Check spelling “or”: do you mean “on”?
  8. Line 114: Poor grammar:  “The hypothesis being that…”

Why didn’t your results match with the literature findings that authoritative parenting style fosters positive SR compared to authoritarian or permissive parenting style?  

Why was parenting not used as an independent variable instead of a moderating variable? Would the results have differed?

Unfortunately, the awkward writing style suffers significantly from lack of editing, particularly the three types of redundancy.  To be precise: be concise. Further problems with writing style included poor sentence structure, lengthy sentences, hyperbole, mismatched phrases, run-on sentences, and occasional spelling errors.   The tedious sentences interfered so much with reading: tiresome and undermine the points the authors are trying to establish. 

Reviewer 2 Report

Self-regulation is a very important competency for preschoolers, and this study employs a strong data base to examine the predictors of this skill.  Their review of the research, statistical analysis, and discussion are all appropriate, and I appreciated their decision to examine the components of self-regulation as well as the construct as a whole..

I found the tables and figures in the paper to be useful but each could be improved.

Figure 1 does not indicate clearly which variables moderate which risk factors.  Figure 2 is excellent, though.

If the variables in Table 1 and Table 2 were flush left, these tables would be much more clear.  Also, in Table 2, it would be helpful to make it visually clear which variables are statistically significant.  Personally, I would leave the interaction terms in the table and clearly indicate with bold font and/or "stars" which variables are significant and which were dropped.  Alternatively, dropped variables might be listed in a footnote.  In the title, note "AOR" after "Adjusted Odds Ratios because tables should be able to stand alone.

As the authors note, several of their variables are pretty thin, in particular the measures of child care and screen time.  For child care, the lack of a quality measure is critical but it is also the case that 10 hours a week is a pretty low cut-off.  A robustness check would assess a higher cut-off.  The screen time variable is what it is, but the discussion might reflect on this variable in an era of covid.

I think that maternal education (or best-educated parent's schooling) should be among the control variables.

Missing data is said to be minor, but this reader needed more explanation of the 58.4% of 2819 eligibles; I assume that this is attrition.  Because it is so high, readers need more information about who was lost. 

Are there no policy or program comments that the authors would find appropriate? 
